# Raising the Green Roof: Enhancing Youth Water Literacy through Built Environment Education

Laura B. Cole [1,*], Lilian Priscilla [2], Laura Zangori [3], Beth Kania-Gosche [4] and Joel Burken [5]

1 Department of Design & Merchandising, Colorado State University, Fort Collins, CO 80523, USA
2 Department of Architectural Studies, University of Missouri, Columbia, MO 65211, USA; lpd5f@mail.missouri.edu
3 Department of Learning, Teaching, & Curriculum, University of Missouri, Columbia, MO 65211, USA; zangoril@missouri.edu
4 Department of Education, Missouri University of Science & Technology, Rolla, MO 65409, USA; bkaniagosche@mst.edu
5 Department of Civil, Architectural and Environmental Engineering, Missouri University of Science & Technology, Rolla, MO 65409, USA; burken@mst.edu
* Correspondence: laura.cole@colostate.edu

**Abstract:** Green roofs cool cities, clean the air, provide habitat, and manage stormwater. They are compelling tools to teach interconnected human-ecological systems. This study included the design, pilot, and evaluation of a fourth-grade science unit entitled "Raising the Green Roof", exploring these connections. Five classrooms in two Midwestern U.S. public elementary schools participated, and 4th-grade students (*n* = 73) drew systems models at three time points (212 models) and wrote their ideas. Qualitative content analyses of the models showed that learners were increasingly combining social systems (green roof infrastructure) with ecological systems (water cycle) across the unit. Students also increasingly evidenced specific knowledge as they progressed through the unit. The analysis of student models revealed that most student confusion is related to built environment aspects (e.g., how water moves from building roofs to municipal waterways). Results of the study suggest the potential for teaching socio-hydrologic systems thinking at the fourth-grade level. The findings emphasize the need to enhance built environment education for youth in science units that aspire to connect features of the built environment, such as green roofs, with ecology. The study additionally reinforced the effectiveness of place-based units in elementary education that emphasize science practices.

**Keywords:** green roof technologies; water literacy; elementary education; socio-hydrologic systems thinking; scientific models

## 1. Introduction

Water scarcity and water cleanliness are global concerns [1]. The way we build our cities profoundly affects natural water cycles, potable water use, and overall water quality as water returns from built to natural environments [2]. Roof design is one built environment factor that has far-reaching consequences for the water cycle in cities. Roofs affect how and where water flows—and the quality of water as it moves from cities back into water tables. To solve multiple problems at once, cities are increasingly looking to green roof technologies. They can be a cost-effective way to mitigate urban flood risk [3,4]. Green roofs—roofs where vegetation is placed over water-proofing—can maximize stormwater retention, thereby reducing runoff and the burden on aging infrastructure and decreasing the volume of water rapidly sent to nearby waterways (through drainage pipes, sewers, etc.) in storm events. Green roofs provide other benefits to the urban environment, such as natural habitat, increased roof life, urban heat island mitigation, reduced air pollution, and reduced energy demands [5]. Intensive green roofs additionally have food production potential, providing

a rich example of a design strategy that contributes to the food-energy-water-ecosystem (FEWE) [6].

Given the numerous social and ecological benefits of green roofs, they have strong potential to be teaching tools for youth to better understand the impacts of human activity on natural systems [7]. The current project converges science and environmental education [8] to illuminate how human choices such as roof design impact natural processes like the water cycle. It falls under broader initiatives to promote green building literacy [9] by using green building science to enhance science education. Our interdisciplinary team designed, implemented and evaluated a place-based water literacy unit based on green roof technologies. This unit was aligned with U.S. national science education standards (Next Generation Science Standards [NGSS]) [7]. Of the many benefits achieved by green roof technologies [5], the focus of this unit was on managing stormwater. Isolating this benefit afforded the opportunity to deeply examine the interconnections between green roofs and the water cycle in a time-limited elementary science unit.

This research study examined if and how fourth-grade students in two U.S. Midwestern public elementary schools developed an increased understanding of the relationships between natural and built environments across the 4-week unit. By examining elementary students' system-thinking abilities, the work here adds to the growing evidence that systems thinking can be integrated into elementary education despite conceptualizations of systems thinking as content more suitable for secondary education [10]. Results of this study suggest that, while students showed an increasing ability to cross socio-ecological systems, the built environment learning content presented challenges to teachers and students alike who had not previously examined the role of the built environment in natural processes such as the water cycle.

### 1.1. Conceptual Framework

The conceptual framework for this study, described in the sections to follow, begins with the broad framing of "water literacy" [11]. We highlight how this framework can be used to cross socio-ecological systems of green roof technologies (part of social systems) with the water cycle (ecosystem process). Our adapted framework shows overlapping systems and, building on the broader water literacy framework, includes a more explicit focus on systems thinking for elementary students, where we examine "socio-hydrologic" systems thinking and employ model-based learning (MBL) as the key learning theory.

### 1.1.1. Water Literacy

Global efforts exist to increase public understanding of water resources and aim to reduce water consumption, increase environmental sustainability, and improve public health [12]. These foundations can begin in the elementary classroom where students recognize the powerful relationship between the water cycle, a staple of the elementary science curriculum, and societal systems, such as life in and around buildings. This intersection of content prepares the next generation to contribute to water-related decision-making in all aspects of human activity [13]. Understanding these relationships is considered water literacy or watershed literacy. McCarroll and Hamann [11] offered a thorough review of these concepts to define water literacy as "the culmination of water-related knowledge, attitudes, and behaviors" [11] (p. 1), which aligns with similar frameworks under the broader umbrella of environmental literacy [14].

The McCarroll & Hamann (2020) literature review was synthesized in a framework that begins with general knowledge in the outer ring and moves to attitudes, values, and individual/collective action in the center. General knowledge refers to simplicity and conciseness, including "basic watershed concepts" and "necessary water knowledge" [11]. Specific knowledge refers to the three-part knowledge portion of the framework that outlines three key specific cognitive domains: (1) Science and Systems Knowledge, (2) Local Knowledge, and (3) Hydrosocial Knowledge. According to McCarroll and Hamann [11], Science and System Knowledge refers to water's scientific properties and its significance for

living systems; Local Knowledge refers to understanding local water sources, infrastructure, water demand, and uses; and Hydrosocial Knowledge refers to continuous interdependency between society and water resources [11] (p. 8).

1.1.2. Socio-Hydrologic Systems Thinking for Elementary Education

The nexus of water cycle content and green roof technologies can be mapped onto the McCarroll & Hamann Water Literacy Framework [11] to show how built environment themes can promote youth water literacy (Figure 1). This adapted framework is the theoretical grounding for this study. We employ the Water Literacy Framework knowledge types [11] to draw connections between the overlapping knowledge domains of science and systems knowledge (water cycle and watershed knowledge), local knowledge (green roof as infrastructure), and human well-being (hydrosocial knowledge). The built environment (as a product of social systems) is here distinguished from hydrosocial knowledge, which relates to the numerous benefits of green roof technologies to individuals and communities (e.g., clean air, water filtration, cooling effects). Previous work employs the term "socio-hydrology" to describe this intersection of water systems with human factors [15]. We thus use the term "socio-hydrologic systems thinking" to describe one's ability to think across systems diagrammed in Figure 1. The curricular unit examined in this study was framed by human well-being themes (i.e., flooding in cities), but the primary science content included the water cycle and green roof technologies. The Figure 1 diagram, therefore, uses bubble size to show these two domains as the key systems examined in the current study.

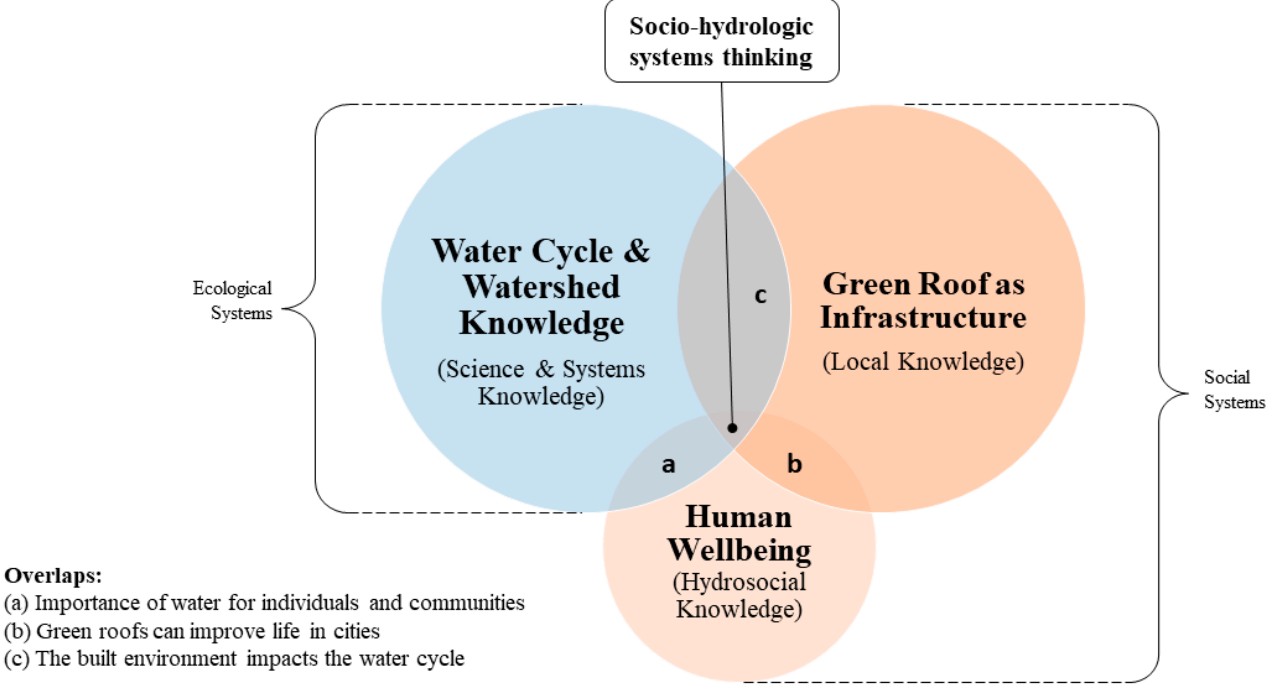

**Figure 1.** Water Literacy Framework adapted for the current study (Source: Authors).

Previous work on water cycle education shows that the social dimension is typically missing in science education. Classroom visual representations of the water cycle have largely been unchanged since the U.S. Geological Survey published the original water cycle diagram in the 1930s [16,17]. This representation tends to highlight only three water cycle processes—evaporation, condensation, and precipitation—and does not explicitly connect societal interrelationships with the water cycle. Prior research has primarily focused on children's knowledge of these three processes [18–20]. This research finds that, overall, youth recognize that water exists in different forms in different places, such as clouds in the atmosphere, in lakes and rivers on the ground, and under soil. When youth

are asked to draw a systems model (a diagram that represents interconnected elements of a system) of the water cycle, they typically do not include other water processes, such as water vapor in the atmosphere, plant transpiration, and/or water moving through the ground and underground storage in aquifers [18]. In addition, youth tend not to place humans and/or societal systems within their water cycle knowledge; they view human activities as separate from the water cycle, and the purpose of the water cycle is for human activities [16,20–22].

While systems thinking is ostensibly a high-order skill, scholars have examined the potential to seed systems thinking at the elementary level, challenging the notion that elementary students are unable to grasp elements that cross system boundaries [10]. When youth knowledge is considered within broader hydrological systems, their knowledge is quite complex and includes cause and effect patterns that explain hydrological system behavior [10,21,23,24]. For example, Ben Zvi Assaraf & Orion [10] found that fourth-grade students' understanding of causal relationships between water cycling and other Earth systems, such as the biosphere and atmosphere, become more complex as they learn about hydrological processes [10]. Fourth-grade students consider hydrological processes occurring across space and time, such as how drought affects the ground in which plants die, which eventually affects water vapor in the atmosphere due to loss of transpiration. Fick et al. [23], in their study of fifth and sixth-grade students, found that students consider the cause and effect between water quantity, surface material, and water runoff when specifically scaffolded to consider these inputs and outputs to the water cycle. Levy and Moore Mensah [24] show that situating water cycle learning within the local environment supports sixth-grade students in integrating interrelationships between water cycling processes, such as water runoff, more effectively into their conceptual water cycle models.

While each of these studies considered some elements of human activity within the lessons, such as humans pumping water from wells [10], they did not explicitly focus on the teaching and learning of socio-hydrologic systems knowledge. Overall, very little work has focused on youth (and adults) socio-hydrologic systems knowledge, where instruction is specifically focused on water cycle and human systems interrelationships and how human systems impact the water cycle and vice versa [16,17,25]. Supporting students in building this knowledge is critical as "humans...dominate critical components of the hydrosphere" [16] (p. 533), and human activity has impacted, in some manner, all water cycle processes. By integrating the built environment, a social system, with the water cycle, the current study explores how social framing around the water cycle can enhance outcomes for students.

### 1.1.3. Categorizations of Systems Knowledge

The McCarroll & Hamann [11] water literacy framework includes an outer ring entitled "General/Unspecified Knowledge", where knowledge becomes increasingly specific as content moves toward the inner rings of the diagram. This distinction between general and specific knowledge types—together with our interest in combining content across social and ecological systems—resulted in a framework for classifying student knowledge demonstrations in the current study (Figure 2). This framework distinguishes between student work that shows few system elements and low elaboration (general) versus work that identifies more than three system elements and shows complex understanding (specific). It additionally distinguishes student work evidencing understanding of an individual system (e.g., water or built environment) versus models that showed interrelationships within combined systems (e.g., how the built environment impacts the water cycle).

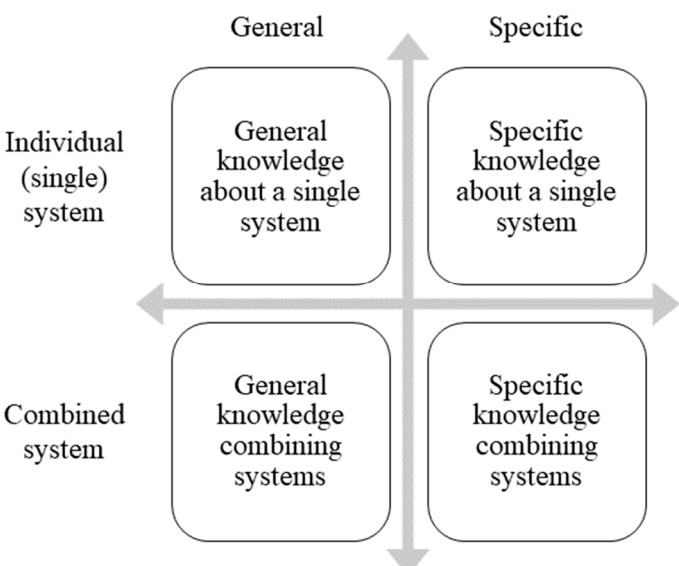

**Figure 2.** Categorizations of Systems Knowledge (Source: Authors).

### 1.1.4. Model-Based Learning

Scientific modeling is identified within the U.S. science standards, titled the Next Generation Science Standards (NGSS) [7], to support K-12 students' development of learning about systems. Models can take many forms (e.g., computational, mathematical, or physical) within the science classroom. Here, we use the modeling form of 2-dimensional (2D) diagrammatic drawings in which each student individually develops their model in response to a question or proposed problem. This modeling approach is considered model-based learning (MBL), as the models provide a conceptual window into student learning. The 2D models that we focus on are considered systems models, defined by the NGSS as models that "define the system under study—specifying its boundaries and making explicit a model of that system" [7] (p. 1). Throughout this paper, we will use the terms "model" or "systems model" to refer to the output of the scientific modeling process.

MBL is situated within cognitivist learning theory traditions [26,27]. It is a multiphase process that occurs through students developing, using, evaluating, and revising their models [22,28,29]. When they initially develop their 2D models, they are accessing what Linn [30] refers to as their preexisting repertoire of models for complex phenomena. This repertoire is developed from prior observations of, experiences with, and inferences about complex systems, such as the water cycle [30]. Students use their developed model as a reasoning tool to describe how and why system processes are happening [31]. To elucidate causal reasoning, we ask students to include what is happening in their model (e.g., green roof plants absorb water) how it is happening (e.g., engineered soil is water-absorbing), and the underlying mechanism for why it is happening (e.g., green roofs help cities manage stormwater) [32]. The focus on what, how, and why defines the explanatory power of the learner's model [33,34]. Explanatory power is how well students are able to articulate the relationships between system elements and variables to each other [7]. Initially, within science education, there was debate as to whether or not primary school students have the ability to causally reason and attribute underlying mechanisms to their models [35]. However, more recent research finds that across growth and development from childhood to adulthood, causal reasoning abilities remain equivalent; what changes is the amount of domain-specific knowledge (knowledge of a given subject matter) available to the individual to use for causal reasoning about complex system behavior [36].

This growth in domain-specific knowledge is visible as students evaluate and revise their systems models. Over the learning experiences, as students gain new knowledge, experiences, and observations, they return to the original model to evaluate its explanatory power. As their repertoire of ideas increases, their causal reasoning also gains in complexity

as they integrate new models into their existing repertoire [30]. However, their original repertoire of conceptual models is not replaced or removed; rather, it is cued less due to its limited power to explain the phenomenon. New ideas with greater explanatory power gain cueing priority [37] as they provide more robust explanations of complex systems. Through evaluating their model, students can reflect on their repertoire of models and revise their model to make their new knowledge visible. Each phase of the modeling process (develop, use, evaluate, and revise) is an opportunity for students to reflect upon and make their knowledge and reasoning visible and explicit. As such, models serve as both representations and tools for students' knowledge-building about complex systems [22]. Furthermore, expressed models serve as a physical artifact of their knowledge as well as a historical artifact of knowledge progression across the learning experience [27].

### 1.1.5. MBL for Socio-Hydrologic Systems Thinking

In summary, with MBL as the underlying learning theory, the current work examines the ability of upper elementary students to think across systems (built environment, human well-being, and hydrologic cycles), continually building their repertoire of ideas with iterative systems models. Figure 3 demonstrates the idealized process for a learner who is experiencing a curricular unit driven by the scientific practice of modeling, where elements of multiple systems are added to models across the unit. The models play a key role in helping learners visualize their thinking, and, in an ideal scenario, the final model drawings exhibit the ways in which learners are combining elements of various social and ecological systems into one model.

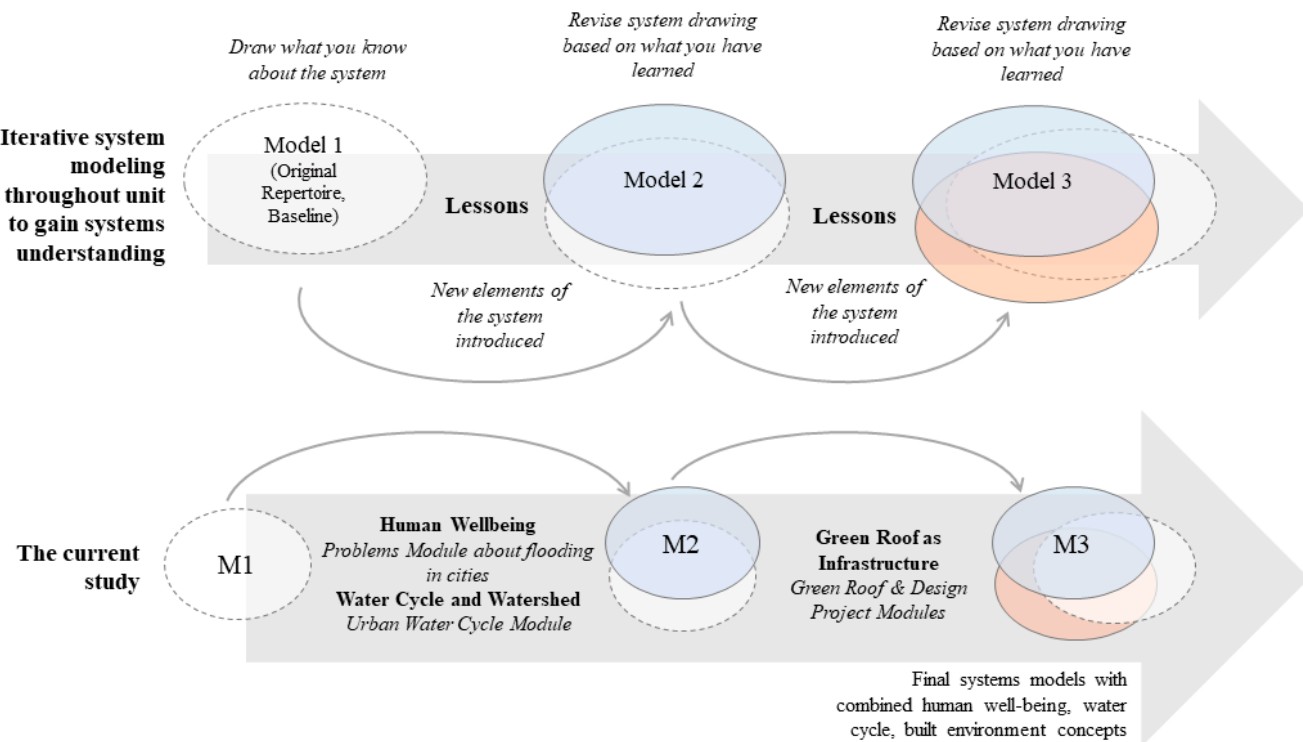

**Figure 3.** Idealized learning progression using models to promote socio-hydrologic systems thinking.

### 2. Materials and Methods

This study examined if and how fourth-grade students increased their ability to communicate the complex interrelationships between the built environment (roofs) and the water cycle across a 4-week unit. The central research questions guiding this work were: Did, and how did, students in the "Raising the Green Roof" unit use MBL to demonstrate socio-hydrologic systems thinking that combined understandings of the water cycle and green roof technologies? To build toward the central research question, the following

questions guided the initial stages of analysis: (1) what science content is represented in the models over time? and (2) did, and how did, the models demonstrate increasingly specific ideas over time? The initial analytical stages uncovered how students identify and refine the components of the system and supported the subsequent analysis by examining how students then link systems together. Our final research question emerged as we engaged in depth with the data: What kind of confusion was expressed in the student models that signal content areas where students required more support? A case study approach [38] was employed with two elementary schools within the same school district where the unit was implemented across five classrooms. Given the focus on conceptual understanding of water flow across human and ecological systems, systems modeling served as an analytical tool to examine student learning outcomes about the interconnections between elements of natural and human systems. The study was awarded ethics approval from the Institutional Review Board (the University of Missouri Institutional Research Board (IRB) project #20432442) prior to beginning data collection activities.

## 2.1. Research Participants

This study takes place in two public elementary schools from the same school district in a suburban town with a population of 55,000 near a larger Midwestern urban center. The district had a sustainability coordinator at the time this study began but no formal sustainability education programming at the elementary level. The Midwestern state where this project occurred uses the Next Generation Science Standards (NGSS) as the basis for state standards in science education. The science coordinator for the district was consulted in the initial stages of unit development and confirmed that the students in our study would have experienced a basic weather and climate water cycle module in third grade that followed the 5E format (Engage, Explore, Explain, Elaborate, and Evaluate) focusing on evaporation, condensation, and precipitation. The district was chosen as a natural audience for the type of environmental education lessons used in this study. The staff at the district level recommended the elementary schools that participated in this study. The schools were chosen to expand our observations across multiple case study schools and enhance the transferability of our results to new settings [38]. Pertinent to the concept of stormwater management, both schools were in a suburban setting with plentiful vegetation across the school campus and local community. This suburb is located approximately 14 miles away from an urban center that contains less vegetation and mid- to high-rise buildings. We use the names Clover Elementary and Bluestem Elementary for the purposes of distinguishing the two schools where needed. In the year of this study, the Clover Elementary student population was 77% White, 7% Black, 7% Multi/Other, 5% Hispanic, and 4% Asian/Pacific Islander, with 3% of students as English Language Learners (ELL). The student population at Bluestem Elementary was 44% White, 32% Hispanic, 14% Black, 7% Multi/Other, 2% Asian/Pacific Islander, and 21% ELL. The percentage of students with free and reduced lunch is 10% at Clover Elementary compared to 56% at Bluestem Elementary. Pseudonyms for research participants are used in reporting.

## 2.2. Raising the Green Roof Water Literacy Unit

The 4-week unit entitled "Raising the Green Roof" was collaboratively developed by an interdisciplinary research team with five elementary educators at two elementary public schools in the U.S. Midwest (two teachers at Bluestem and three teachers at Clover). Prior to implementation, educators and researchers convened in a professional development workshop to test and improve the lessons for use in the fourth-grade science classroom. The resultant unit featured four modules containing experiential learning activities, place-based learning activities, and multimedia presentations. The modules were: (1) stormwater problems in the built environment, (2) the urban water cycle, (3) green roofs, and (4) the application of learning to a green roof doghouse design project (Figure 4).

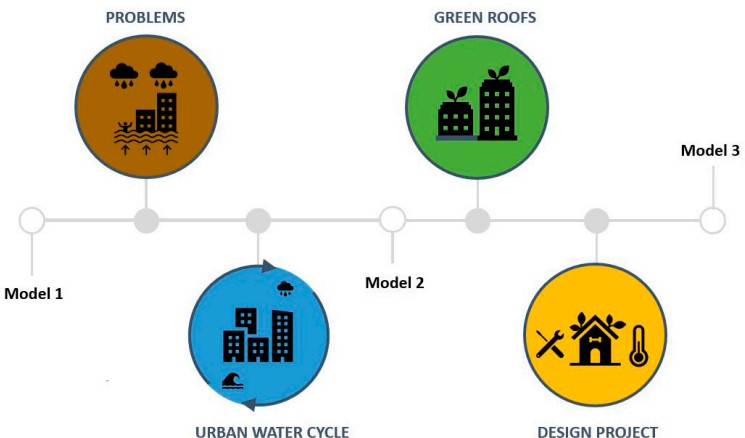

**Figure 4.** Unit Timeline (Source: Authors).

The "Problems" module provided a framework for human well-being for the unit by forefronting the ways that poor water management in cities affects people. This module contained numerous place-based lessons to help students make a personal connection to water in their own urban areas. Examples of significant floods in the nearby river were used to show how rain events affect life in cities. Students examined Google maps of their own cities, comparing hard concrete surfaces to vegetated areas. In this module, students explored water flow on their own school grounds in an activity called "Rain to Drain", where learners were able to explore how rain hits the roof of their school building and travels toward city systems. The "Urban Water Cycle" module helped students revisit water cycle elements they likely learned in previous grades (e.g., evaporation, condensation, precipitation) and build on this knowledge to connect ideas to urban areas. Students conducted hands-on water cycle lessons and compared the water cycle in urban to more natural, vegetated areas. They worked hands-on with native plants to understand plant transpiration. The third module on "Green Roofs" gave learners a more in-depth understanding of what a green roof is, how it is made (e.g., the different layers that promote water absorption), and how it interacts overall with the water cycle. The COVID-19 pandemic precluded a field trip; however, students experienced a live presentation from an engineer broadcasting from a green roof. The last activity in this module gave students the opportunity to create their own miniature green roof using a shoebox. The final "Design Project" invited students to apply their knowledge in an abbreviated engineering design process to design a doghouse with a green roof. Students were given a Siberian Husky named Luna as their client and challenged to design her new home by applying lessons learned in the unit. See Supplementary Materials (Figure S1) for photos of lesson enactment (at a pilot school that is not represented in data for the current study).

*2.3. Data Collection*

Within the case study research design, a variety of data were collected for this study. While the main data sources for the current study were the student modeling packets (*n* = 73, 212 models), teacher focus groups, and weekly surveys (*n* = 5) were additionally collected to support the qualitative data analysis and emerging interpretations of model data.

Students developed three 2D pencil-and-paper systems models to the question, "How does a building's roof connect to the water cycle?" Students wrote about what their model shows and how their roof affects the water cycle and answered why this was important. Models were drawn at three points in time (pre-, mid-, and post-unit, as indicated in Figure 4) using the same prompt at each time point. Data collection followed procedures used by previous scholars who have adopted 2D diagrammatic models in research [27,29,39,40]. For each time point, students were given about 20 min to draw their model and answer the four reflective questions following. These questions asked students to articulate the what,

how, and why within their model drawings. During Models 2 and 3, students were asked to evaluate their previous model by rating their previous model and explaining the rating. They were additionally asked to reflect on what they had learned since the last model drawings (see File S1 modeling packet in Supplementary Materials). Student drawings and writings were collected in packets by teachers and scanned for the research team at the end of the unit.

Teacher focus groups were conducted to gain the educators' perspectives on the curricular materials and experiences enacting the materials in the classroom. The first focus group occurred during the teacher professional development workshop in which educators ($n = 5$) tested and helped to improve the curricular materials in advance of using the materials. The subsequent focus groups included only educators at Clover Elementary ($n = 3$), who joined an in-person focus group at the end of the unit and then joined the research team again six months later for a member-checking session to help the research team better interpret emerging results. Bluestem Elementary educators expressed time constraints that made additional meetings challenging to schedule. All teachers across schools, however, participated in a weekly teacher survey. This short online survey was administered every Friday during unit enactment, and teachers were asked to list the lessons they completed that week and share successes, challenges, and adaptations needed.

*2.4. Data Analysis*

All data were scanned, transcribed, and imported into Dedoose (v9.2.007) analytical software. Prior to importing the student 2D models, all writing completed for the question prompts was typed into data displays that included the student drawing along with writing, all organized by time-point. Each student model, therefore, is a unit of analysis that encapsulates both drawings and writings. The systems models were completed pre- (Model 1), mid- (Model 2), and post- (Model 3) (Figure 4), which resulted in 212 model drawings across 73 students and three time points. The data are thus visual and verbal in nature and contains sequential progress. Seven students were absent on days that modeling activities occurred and only completed two models. They were included in analyses of model trends, but if they missed Model 1 or 3, they were excluded in the analyses described below that involved a comparison of pre/post models. Sixty-nine students completed both Models 1 and 3.

The multi-phase qualitative coding process is diagrammed in the Supplemental Materials (Figure S2). Before we could address our central research question regarding systems thinking, we first sought to identify the elements included in student models and the level of generality versus specificity of the knowledge demonstrated. The analysis of models thus began with open coding to identify patterns grounded in the data [41], which allowed for the emergence of both expected and unexpected patterns. We then used the Figure 2 water literacy framework adapted from McCarroll and Hamann [11] to group codes into larger themes of (1) water cycle knowledge, (2) built environment knowledge, and (3) combined water cycle and built environment knowledge [11]. After grouping codes into these three themes, and to better understand the depth of student thinking, we used the McCarroll and Hamann [11] framework to divide codes into "general unspecified knowledge" versus "specific knowledge".

While the general versus specific categorizations show trends in how students sharpened content knowledge across the unit, this analysis did not yet illuminate our central research question on socio-hydrologic systems thinking. To better understand how and if students were combining socio-hydrologic systems in their models, we examined the changes to student models over time. This process involved a review of all three models together for each student. Model 1 was compared to Model 2 and Model 3 to examine the potential demonstrated movement in student knowledge across the unit. Students provided written reflections on their performance on the previous models (see File S1 student modeling packet in Supplementary Materials) that were used during analysis to enhance researcher interpretations of changes in student understanding. This analysis led

to the categorization of each student model as either general or specific, and either individual or combined, such that each model had a designation such as "general combined" or "individual specific" falling into one of Figure 2 quadrants (as described in Section 1.1.3).

The result of this process was a database that included each student model as having a binary 1 (yes) or 0 (no) for general or specific and individual or combined systems. The student in a given time point was the unit of analysis, where the student was counted within a category if they had at least one portion of their model coded in that category. Codes were only counted once per student per model, even if the concept emerged numerous times across the same model, which ensured that our results were not driven by high-performing students. After models at all time points (Models 1–3) had been coded, the research team conducted inter-rater reliability tests with a researcher outside the project team. This outside coder rated four students in each of the five case study classrooms (20 students, 27% of the sample). After three rounds of coding and consensus-building, the coders reached a raw agreement of 88% and a Cohen's kappa indicating substantial agreement (0.715, $p < 0.001$) [42]. The final coding rubric based on consensus is shown in Table 1.

**Table 1.** Systems Model Coding Rubric.

| Research Question Guiding Analysis | Category | Description |
|---|---|---|
| Did, and how did, the models demonstrate increasingly specific ideas by the end of the unit? | General Knowledge | The writing/drawing does not elaborate on more than 3 distinct elements featured in the unit (<4 elements identified from either water cycle, built environment, or both). Overall presentation of system elements is general in nature and elements are under-described or lacking precision in the drawing, writing, or both. |
| | Specific Knowledge | The writing/drawing elaborates on more than 3 distinct elements (has 4+ elements from the unit identified from either water cycle, built environment, or both). Student shows a level of detail that advances beyond generalities, using scientific terminology (e.g., precipitation versus rain). Models that show understanding of technical concepts (without precise terminology) can also be considered specific if the student is clearly showing a conceptual understanding of a scientific process. |
| Did, and how did, students demonstrate socio-hydrologic systems thinking that combined understandings of the water cycle and green roof technologies? | Individual (single) System | The model drawing/writing does not show a clear connection between the built environment and how it impacts the water cycle (i.e., the student may have drawn both, but the interrelationship between the two is not articulated). This often shows up as a clear emphasis on one system, such as a student focusing strongly on the building elements or water cycle elements, but not managing to clearly connect the two. |
| | Combined Systems | The model activity prompted students to draw an answer to the question: "How does a building's roof connect to the water cycle?" Combined models need to show that students are connecting, or approaching a connection between, socio-hydrologic systems. The model drawing/writing clearly shows at least one way that the built environment impacts the water cycle. This can be demonstrated in both words and drawings. Showing a green roof alone is not sufficient to be categorized as a "combined systems" model. Student needs to show the roof in connection with the water cycle and/or articulate that connection in their writings. |

Once coding agreement was established, the final database allowed the team to generate descriptive statistics that further elucidate patterns in the data over time, showing percentages of models that fell into the various categories across Models 1–3. This database also allowed for the statistical comparison between matched data from pre (Model 1) to post (Model 3) using the McNemar test, which determines statistical differences on a dichotomous dependent variable. We used this test with the "specific" and "combined" model counts for Model 1 versus 3 (given that "general" and "specific" columns would yield duplicate results).

As the research team conducted Phase 1–2 coding described above, patterns of inaccuracies became apparent and catalyzed a final round of coding. This resulted in a final research question for our study regarding student confusion and required a third analysis that focused on model inaccuracies. For this analysis, one researcher flagged initial instances during knowledge progression analysis, and a second researcher did a thorough content analysis of all the model data to expand and develop thematic categories.

All model analyses were complemented by pre and post-focus group data with the teaching team (*n* = 5), teacher weekly surveys (*n* = 5), and a teacher focus group with member checking six months after unit implementation (*n* = 3). The transcriptions for the focus groups, along with the survey outputs, were coded in Dedoose using an analytical procedure that was grounded in the data [43] but also sequentially following the model analysis, which allowed the research team to use secondary data to assist with interpretations of the student model findings. Insights from across teacher data sources were collected into research memos that provided the basis for integrating teacher findings into the reporting of results [43].

All qualitative content analyses were conducted using a collaborative consensus-based process. The graduate assistants on the team were given pertinent readings in advance and trained by the principal investigator on the coding techniques and software employed. To enhance the reliability of the analyses, we used "dialogical intersubjectivity" and simple group consensus as a goal for agreement [44,45]. Two team members coded data and cross-checked each other's work [46], using the consensus-based process to debate and improve codes. These codes were presented to the full team (three additional researchers) at several key checkpoints where debate and discussions of the codes and their categorizations continued. Throughout the process, one team member was the "codebook editor" [47] (p. 132) and was in charge of managing the coding process and recording team decisions. Approximately 6–8 iterations of the codebook were needed to achieve consensus, which is the typical number of cycles in qualitative coding processes [48]. The interdisciplinarity of our team across built environment and science education scholars helped to reduce disciplinary bias in the analysis.

The study had several notable limitations. Due to COVID-19 restrictions at the time of this study, the research team was not permitted to observe lesson enactments or interview students in person. Future research involving this unit would benefit from onsite research activities. Teacher focus groups and weekly online surveys allowed the researchers to follow lesson enactments across the four-week unit and ensure that the lessons were followed in the order suggested and in roughly similar timelines across two schools and five participating teachers. While the study was conducted at two unique public schools with diverse populations, both schools are located in the same Midwestern school district where the largest percentage of students are white. The school with the greatest ethnic diversity (Bluestem) is well represented in the student data, but time constraints on teachers limited the number of touchpoints between teachers and the research team.

## 3. Results

The presentation of results begins with the overarching trends discovered in the student models related to the demonstrations of both general and specific types of knowledge across water cycle and built environment concepts. We then present a more in-depth view of the ways in which different students demonstrated their knowledge across the unit. We then summarize the key confusions coded on student drawings.

### 3.1. What Science Content Is Represented in Models over Time?

The analysis of models over time using the McCarroll and Hamann [11] concepts of general versus specific knowledge shows that "general unspecified knowledge" decreased over time while student "specific knowledge" about green roofs and the water cycle increased. The codes were additionally grouped by the components of the Figure 3 diagram (water cycle, built environment, and the intersection of the two). The codes that were

grouped into these overarching categories are shown in Table 2. Figure 5a shows how vague, general knowledge about the water cycle and/or the built environment all decreased from Models 1 to 3. These codes included students' ability to identify water cycle elements such as rain or broad comments about drains that show general built environment knowledge.

**Table 2.** General and Specific Knowledge Codes from student models (*n* = 73, 212 models).

| | Water Cycle & Watershed Concepts (Individual System) | The Built Environment Impacts the Water Cycle (Combined Systems Knowledge) | Built Environment Knowledge of Green Roof (Individual System) |
|---|---|---|---|
| General Un-specified Knowledge | Identified water cycle elements (71)<br>Rain identified as water cycle element (31) | Drainage elements connected to water cycle (53)<br>Built environment affects water cycle (6)<br>Green roof helps water cycle (6)<br>Roof is good for water cycle (3) | Identified drainage elements (72)<br>Green roof is good for the environment (7) |
| Specific Knowledge | **Science & System Knowledge**<br><br><br>Understand water cycle process (97)<br>Plants are part of the water cycle (49)<br>Irrigate plants (37)<br>Clean air (8)<br>Watershed understanding (8)<br>Water cycle purifies water (5) | **Science & System Knowledge**<br>Drainage system understanding (60)<br>Green roof retains water (21)<br>Ability of surface to absorb water (20)<br>Water runoff (15)<br>Roof is part of water cycle (13)<br>Roof shape affects water movement (10)<br>Biodiversity in built environment (8)<br>Roof is part of watershed (6)<br>Green roof cleans water (5)<br>Roof surface is impervious (4)<br>Green roof plants create oxygen (4) | Green roof requirements (12)<br>Understand different roof types (12)<br>Building damage (8)<br>Roof Safety concerns (4)<br>Green roof insulates the building (3) |
| | **Hydrosocial Knowledge**<br>Clean water supply (21)<br>Water underlies all human activity (21)<br>Watershed and catchment basin health (17)<br>Stormwater can be processed to clean water (8)<br>Conserve water (6) | **Hydrosocial Knowledge**<br><br><br>Flooding (10)<br>Pollution (7) | |

Note: numbers in parentheses refer to the number of times the code occurred across all models.

While general knowledge code frequencies decreased over time, patterns in the codes indicate that student demonstrations of specific knowledge increased over time (Figure 5b). Figure 5b shows several clear trends. First, demonstrations of water cycle knowledge started high, which was likely due to learning in prior academic years. The water cycle lesson in the current unit was taught between Model 1 and 2, and there is corresponding evidence of specific water cycle knowledge increasing across the models (Figure 5b). The percentage of students clearly articulating specific water cycle knowledge increased from 61% in Model 1 to 84% in Model 3. The green roof technology lesson occurred between Models 2–3, and built environment concept codes increased between Models 2–3 from 9% to 29% of students (Figure 5b). Student complex thinking is operationalized as the codes that demonstrated a clear overlap between buildings and nature, here labeled "Combined Water Cycle with Built Environment" (Figure 5b). The student models showed an increasing ability to articulate the interconnections of these systems. The "specific knowledge" rows in Table 2 show the learning content evidenced across student models. Some examples included the student's ability to articulate how roof shape affects water runoff and/or how roof materiality affects the absorption of water.

The teacher data supported findings from the analysis of the student models. Teachers in the post-unit focus group communicated that the unit was very engaging for students. This was especially the case for the experiential and outdoor activities. One teacher, discussing the "rain to drain" tour of the school grounds, said:

Anytime they saw like a sewer, like a manhole cover, they were like, 'Oh, LOOK! That's a drain, that's a drain!' And then I would say 'well, that one is not a drain, but there could be a drain nearby.' So, then they would look. They were like racing around the building to find all the scuppers and document it all. (Nancy)

The most effective activities, according to teachers, were the solar bowl water cycle demonstration (where muddy water evaporated into vapor and condensed to form clear water droplets), the shoebox where students grew their own small green roof, the rain to drain school tour, measuring plant transpiration, and the culminating engineering design project where students designed a doghouse. These lessons appeared to engage students, giving them tangible ideas on how to improve their model drawings on the water cycle, green roofs, or both.

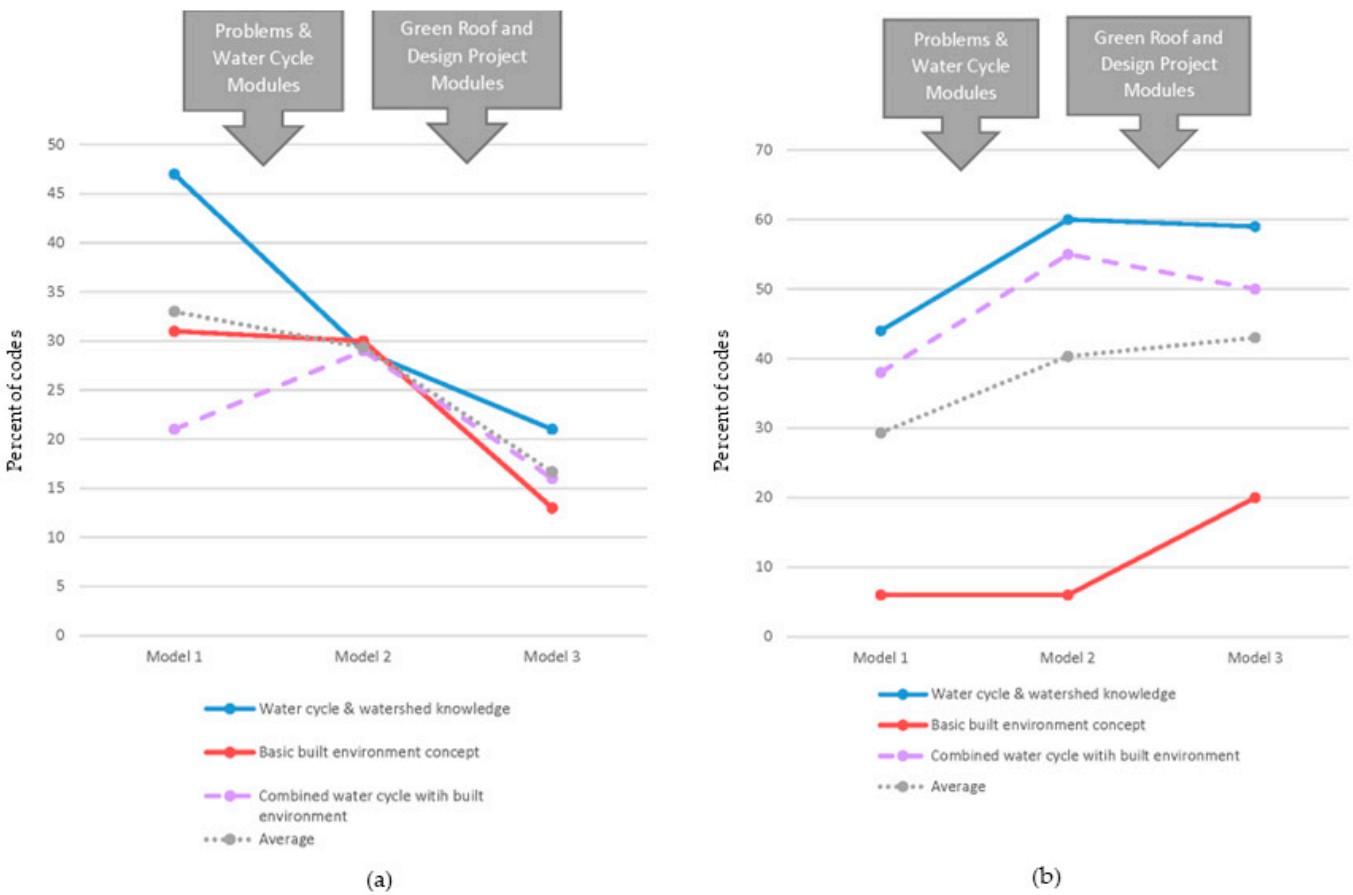

**Figure 5.** Trends in scientific content represented in student models over time by system and split by (**a**) "General Unspecified Knowledge" and (**b**) "Specific Knowledge" ($n = 73$, 212 models).

### 3.2. How Was Socio-Hydrologic Systems Thinking Expressed throughout the Unit?

The second analysis categorizing individual student models ($n = 69$) echoed the first analysis of trends in systems models and showed a dramatic increase in the specificity of models from timepoint 1 to 3, where models coded as "specific" increased significantly from 2.9% to 46.4% ($p < 0.001$) (Figure 6a). Students also increasingly demonstrated "combined systems" in their final models, with a statistically significant increase from 13.0% in Model 1 to 72.5% in Model 3 ($p < 0.001$) (Figure 6b). Figure 7 shows the frequencies of models that were categorized as General Individual, General Combined, Specific Individual, and Specific Combined (Figure 2 framework) from Model 1 to Model 3. These charts show the dominance of General Individual models at the Model 1 timepoint compared to the rise in Specific Individual models at the Model 3 timepoint. The sections to follow elaborate on

the nature of these trends by highlighting student model progressions that were variously categorized within one of the Figure 2 quadrants.

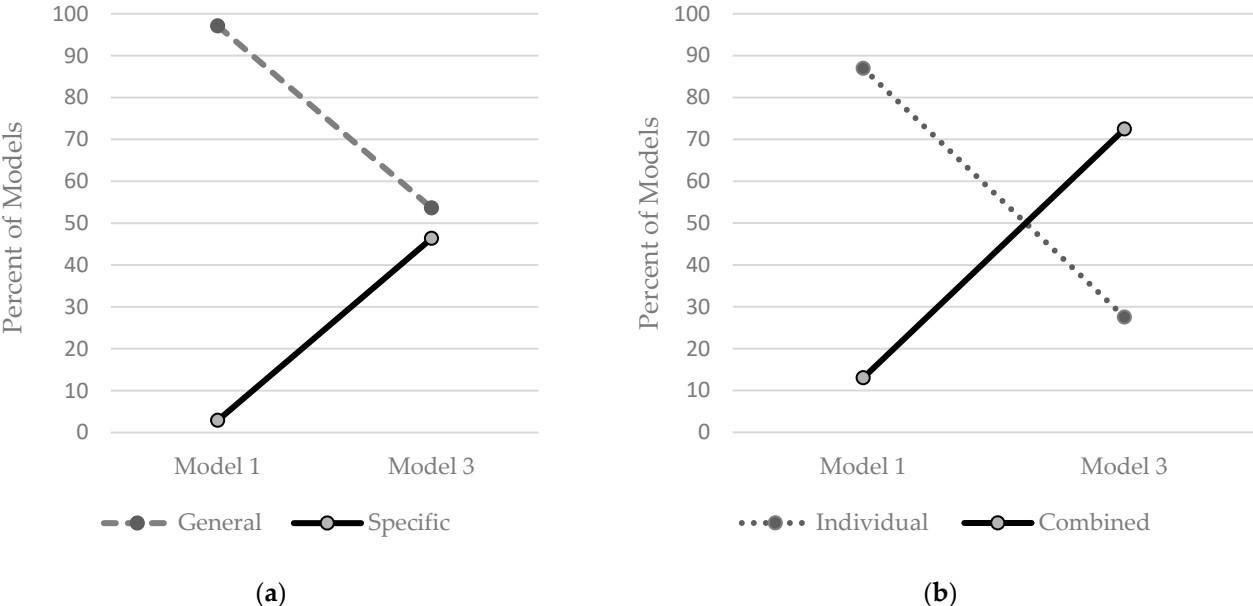

(**a**)                    (**b**)

**Figure 6.** Changes in Socio-Hydrologic Systems Thinking from Model 1 to 3 for (**a**) General vs. Specific and (**b**) Individual vs. Combined (*n* = 69, 138 models).

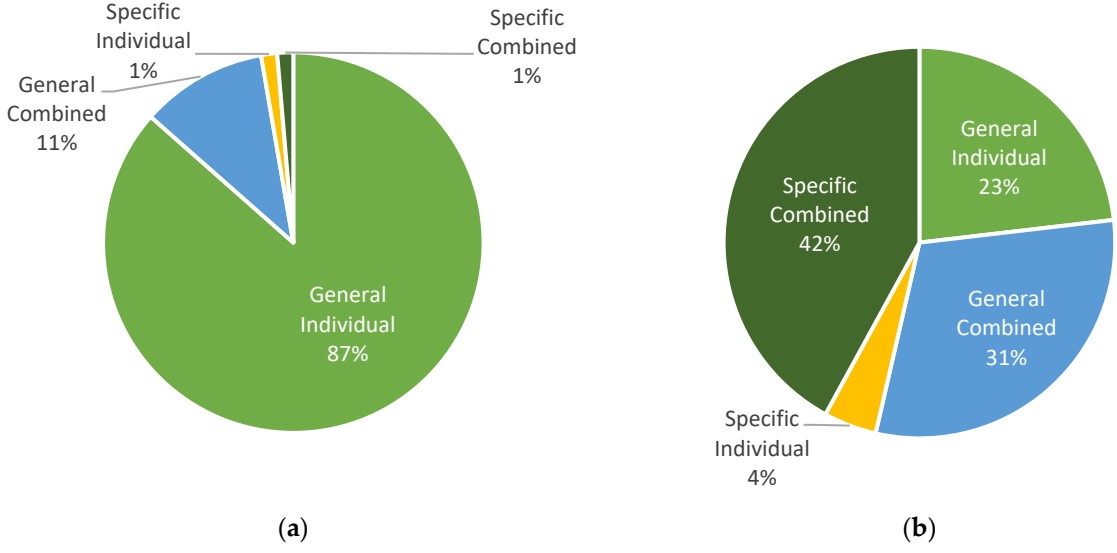

(**a**)                    (**b**)

**Figure 7.** Socio-Hydrologic Systems Thinking reflected in (**a**) Model 1 versus (**b**) Model 3 (*n* = 69, 138 models).

### 3.2.1. General versus Specific

"General" refers to model progressions that showed low overall understanding or identified one or two elements of the systems with minimal elaboration. If further explanations are provided, they typically lack precision or contain one or more factual inaccuracies. For example, in Figure 8 (Model 1), Chris drew a building and drainpipe as a connection. In the second model, he showed an understanding that different roof shapes will improve water flow. He wrote, "some roofs limit water's movement, and others are designed to help water move". In Model 3, water cycle elements (precipitation and evaporation) are present on the roof, but there is no further elaboration on how they interact. He demonstrates a general understanding of the two systems as well as an inaccuracy that water needs help to

move. Chris wrote: "Water needs help moving itself to different places". When prompted to state how his roof affects the water cycle, he said: "it helps water travel quickly to the ground, and it also helps the water cycle". He noted this is important because "the water cycle needs to continue".

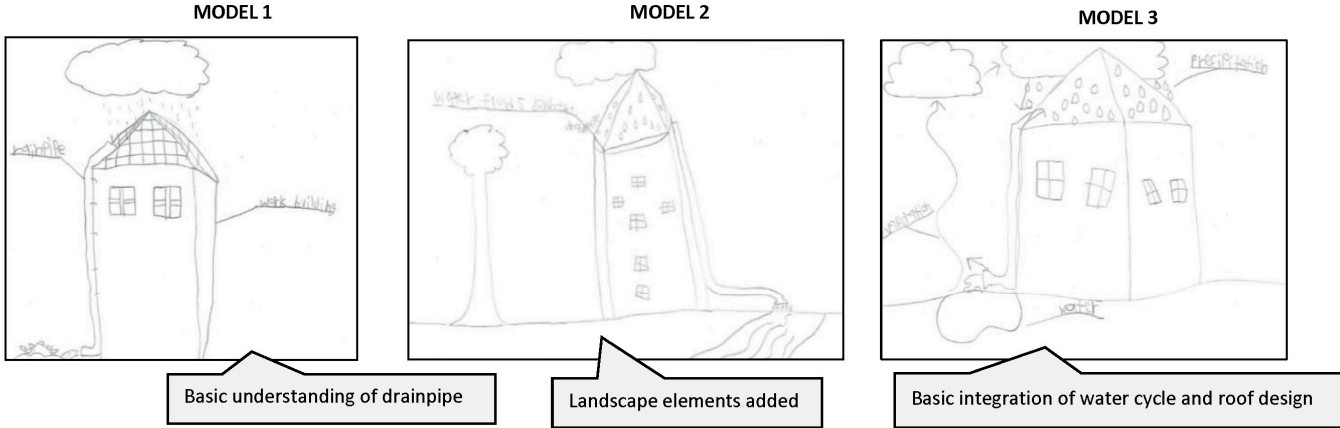

**Figure 8.** Sample "General Combined" systems models (Chris).

"Specific" refers to model progressions that identified more than three elements or characteristics of the systems and demonstrated specific and/or complex understanding with their reasoning articulated. For example, in Figure 9, Liliana displayed prior knowledge of the water cycle and how it interacts with the built environment in Model 1. In Model 2, she demonstrated advanced knowledge of the water cycle, integrating a green roof into her drawing. In Model 3, in addition to these elements, she shows an in-depth understanding of the features of the green roof. This model progression exemplifies the trends toward increasing specificity with the details added as the student gained knowledge of the concepts.

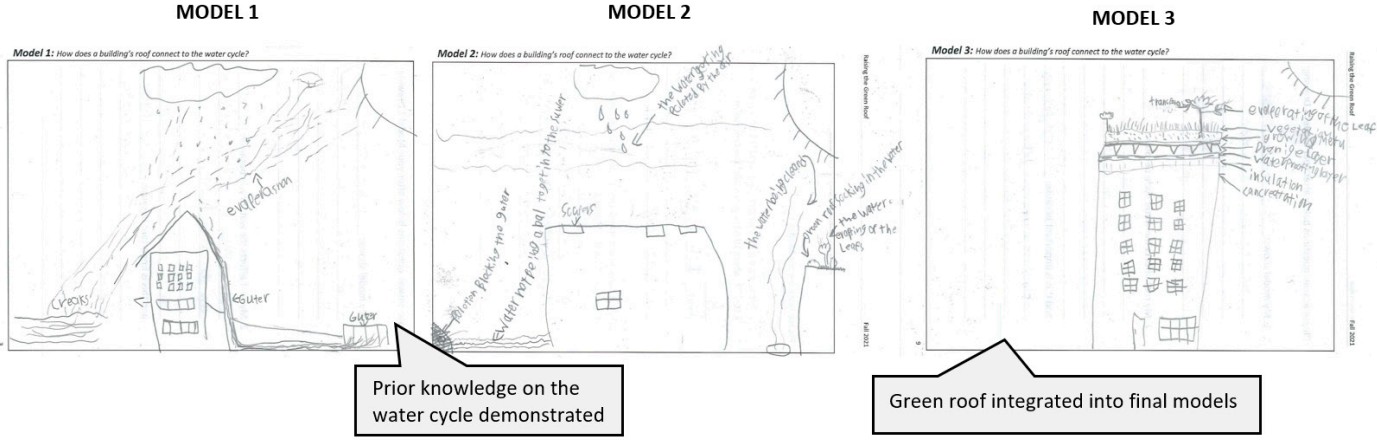

**Figure 9.** Sample "Specific Combined" systems models (Liliana).

### 3.2.2. Individual Systems versus Combined Systems

The next portion of the analysis layers in the evidence for the ways in which students were linking systems or maintaining focus on a single system throughout the unit. Wiley represents students who display individual systems categories with general knowledge (Figure 10). While he demonstrates built environment features and precipitation (unlabeled), the details are few and built environment and the water cycle systems remain disconnected. In Figure 11, Audrey's models show an example of a student who displays individual systems with specific knowledge. Audrey focused on the rain as the water cycle

element in all three models. While water cycle knowledge seems to grow in Model 3 as she noted evaporation and water movement, there is no evidence that she is connecting the water cycle and built environments.

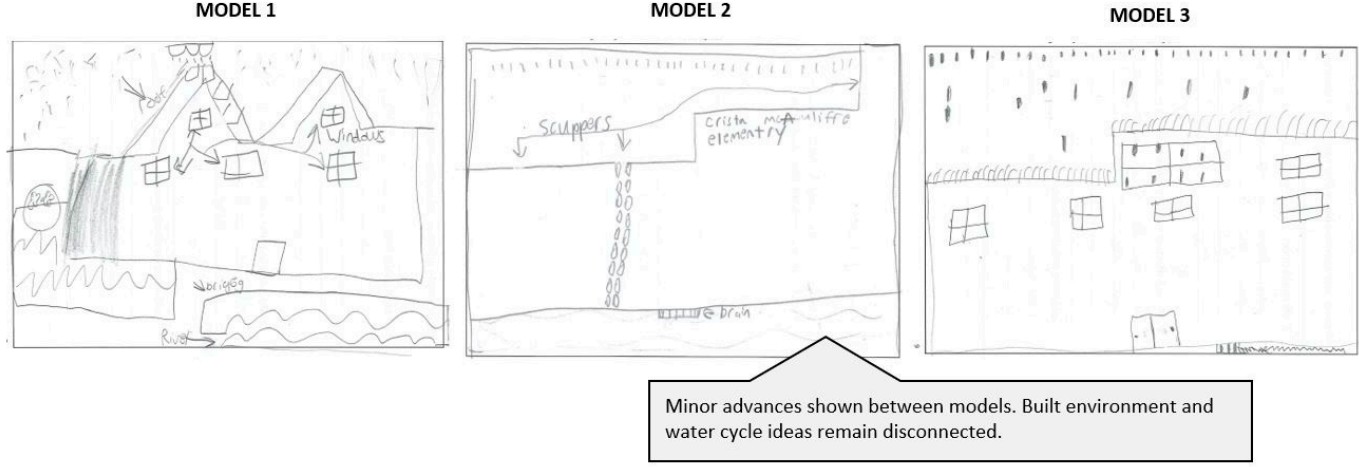

**Figure 10.** Sample "General Individual" systems models (Wiley).

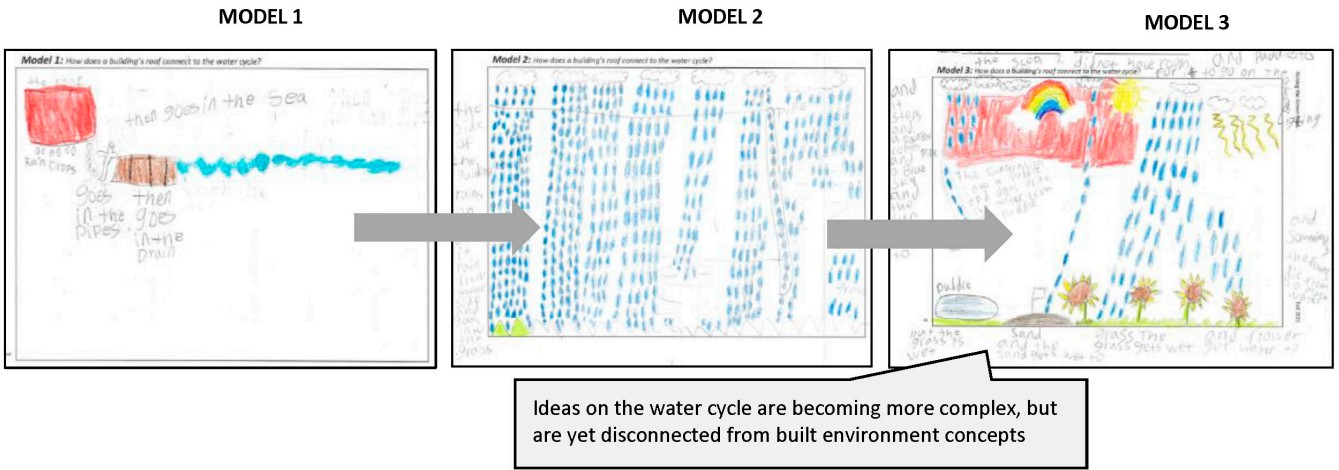

**Figure 11.** Sample "Specific Individual" systems models (Audrey).

An example of combined systems with general knowledge is Hamish (Figure 12). In Models 1 and 2, he drew rain interacting with a house with a roof with an explanation "that water runs off of roofs" in Model 2. Although there is no clear progress between the first two models, in Model 3, he displayed growth in water cycle knowledge and noted that "roofs are part of the water cycle". He made the connection between the two systems, yet the demonstrated knowledge remains general in nature. He mentioned elements of both systems, such as precipitation, condensation, evaporation, water runoff, and green roofs, yet there is no further explanation of the relationship or cause and effect discussed between the systems. Alyssa provides an example of a knowledge progression that combines systems with specific knowledge (Figure 13). Model 1 shows basic roof drainage knowledge, and Model 2 shows an increased understanding of the water cycle. She noted: "I've learned a lot. I learned how the water cycle works and how water evaporates then turns into clouds and drops straight down". In Model 3, she wrote, "...green roofs help the planet by soaking up water and stopping floods". She combined water cycle knowledge with green roof technologies. Another example is Phoebe (Figure 14), who showed prior knowledge of the water cycle in Model 1. She connected the roof and water cycle with the tree life cycle. In Model 2, she demonstrates the understanding that plants on roofs reduce stormwater runoff. In her model, she wrote, "without plants, there would be more flooding, and the

water cycle would be messed up without plants". The third model displays a comparison of the stormwater runoff between a green roof and a regular roof.

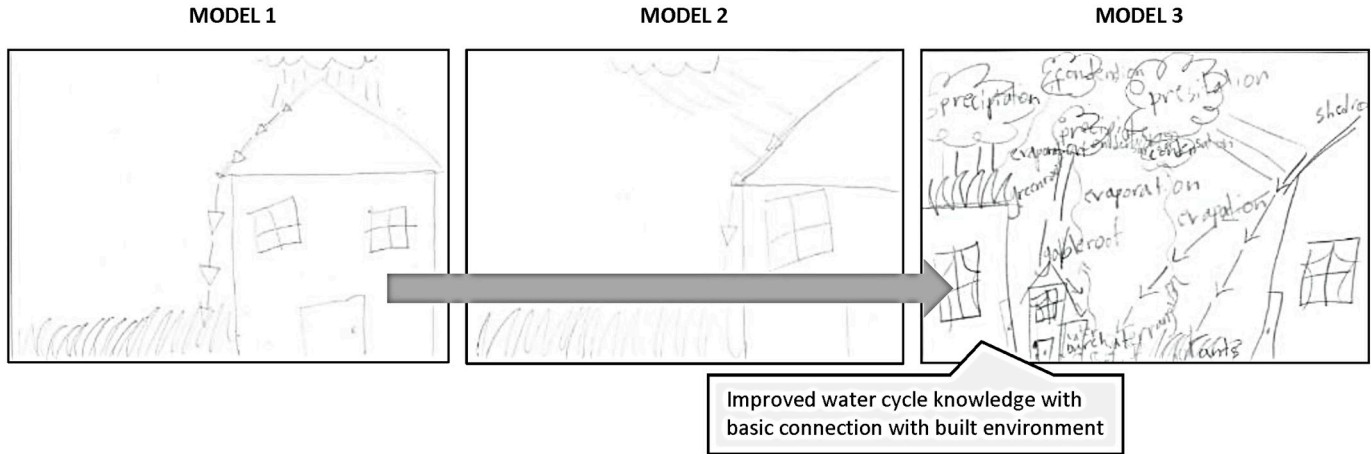

**Figure 12.** Sample "General Individual" systems models (Hamish).

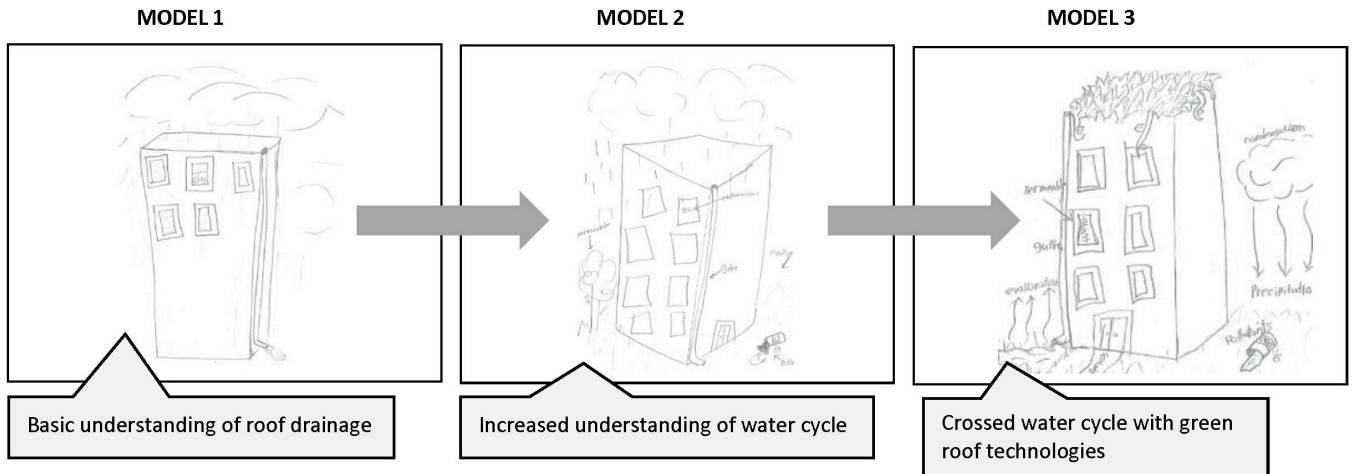

**Figure 13.** Sample "Specific Combined" systems models (Alyssa).

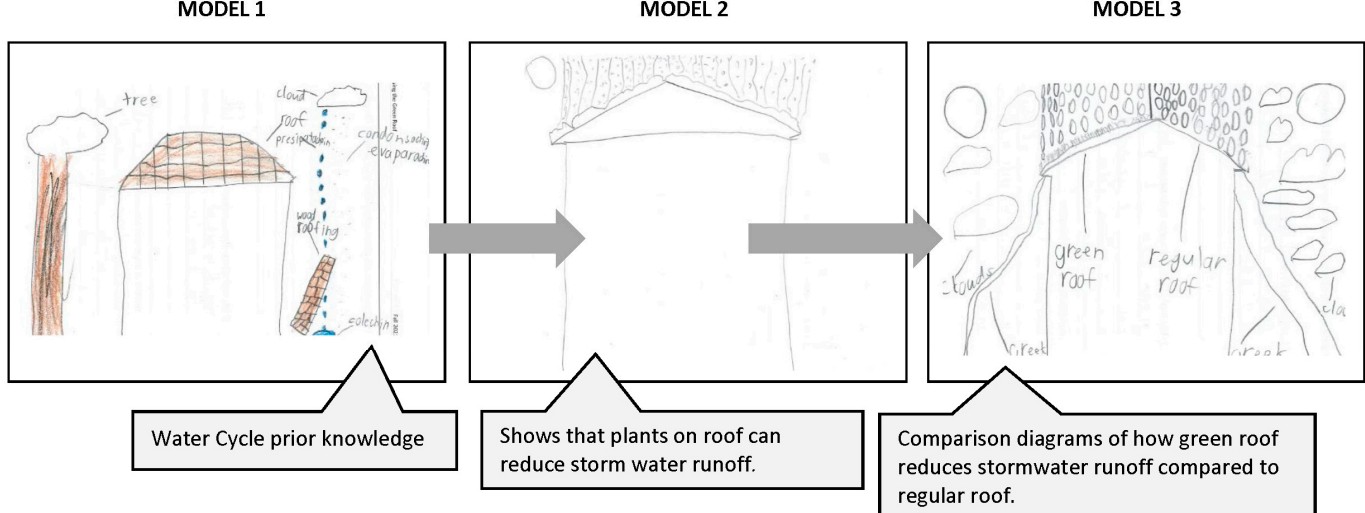

**Figure 14.** Sample "Specific Combined" systems models (Phoebe).

While the analysis of student models revealed that a majority of fourth-graders were demonstrating the ability to combine systems in model drawings (72% in Model 3), their teachers expressed doubts. In the post-unit focus group, Clover Elementary educators were uncertain about their students' abilities to combine green roof and water cycle concepts. Kate, for example, said:

> We talked about the problems and then the water cycle and then the green roof. I just feel like there was still some disconnect, and I don't know how to fix that disconnect. I don't know. But I feel like they still know some of the parts and like they know that the water is cleaned as it goes through [the roof]. (Kate)

Jean agreed and told the research team, "connecting the green roof and the water cycle, I really don't know what you will see". The Clover teaching team, however, did perceive that their students were highly engaged and made important learning gains. It was clear to them that students' water cycle knowledge had increased, and their intuition about that outcome aligns with the student model analysis that showed students increasing their specificity of water cycle elements from pre- to post-unit (Figures 6a and 7). The teachers were additionally confident that student awareness of green roof technologies was heightened, even if students' in-depth understanding of the underlying science—and interconnections with the water cycle—was less obvious to teachers.

### 3.3. What Content Areas in the Unit Needed More Support?

The preceding analyses examined the ways in which students demonstrated their knowledge of socio-hydrologic systems over time. However, a more complete picture of student thinking includes the ways in which systems models were incomplete or demonstrated factual inaccuracies. The models were not presented to students as a test and are not conceptualized as such. Given the open-ended nature of the model prompt, there were many ways to complete this exercise. Our analyses of models were centered on the way that student thinking evolved. In the process, patterns of student confusion illuminated ways in which the "Raising the Green Roof Unit", and similar units, can better support learners in developing complete and accurate systems models of the phenomenon.

Most model inaccuracies were related to the built environment, a finding which aligns with the low demonstration of specific built environment knowledge in student models overall (Figure 5b). The notable patterns among students, beginning with the most prevalent, included: (1) confusing water outside versus inside buildings, (2) omitting key elements such as drainpipes, (3) assuming that rain coming off roofs typically helps plant life below the building, (4) indicating that water sits in pools on roofs before it evaporates, and (5) stating that roofs are necessary to support the water cycle. This fifth theme relates to a sixth we call "epic consequences", where students showed a tendency to overstate or dramatize the importance of the process. Each theme is elaborated below (beginning with the most prevalent themes), and Table 3 provides an overview of the key themes with sample excerpts from student models.

**Table 3.** Student ideas and content areas need more support.

| Student Ideas | Sample Student Writings |
| --- | --- |
| Water outside versus in | "My model shows a green roof on the house and the pipes connect to everything, tub, sink, toilet, also scuppers" (Lucy, Model 3) <br> "My roof affects the water cycle because the pipes bring water in and in the roof". (Tom, Model 1) |
| Missing drainage elements | "I didn't add the plant to the roof and the pipes". (Mason, Model 1 evaluation at Model 2 timepoint) <br> "I learned that water goes in drain and helps the plants and tree roof" (Jackie, Model 1 evaluation at Model 2 timepoint) |

| Student Ideas | Sample Student Writings |
|---|---|
| Roofs water plants below | "The water pipe makes the rain go in this tube to the ground. And this is important because it helps grass grow". (Tiffany, Model 2)<br>"The water goes in the drain and this is important because it helps plant grow". (Joseph, Model 2) |
| Water just hangs out on roofs | "My roof affect water cycle because (water) stays on the roof and it evaporates. This is important because the water cycle so the water doesn't get stuck". (Eve, Model 2)<br>"(The rain) gets stuck on the roof and then if comes to the floor". (Brian, Model 1) |
| The water cycle requires roof | "(Rain water) does not evaporate right away and (the roof) is important because clouds won't come". (Brian, Model 1)<br>"The water goes down the sides and makes puddles which gets evaporated into the sky. And this is important because it helps the flow of rain". (Matthew, Model 1) |
| Epic consequences | "If we didn't have the water cycle, we cannot have clean water and we couldn't live". (Rachel, Model 2)<br>"The roof makes the water go in the (pipe) and makes it go into the water puddle and in the grass and helps plants. If the water did not go into the grass and make puddles, everything would be dead". (Jackie, Model 2) |

### 3.3.1. Water Outside versus in

Student models show numerous confusions about the connection between the water cycle, the roof, and what happens inside buildings. This included comments about rainwater being confused with potable water, water damage that happens to roofs, water leaking into interiors, and so on. Students would interchange rainwater, toilet water, and drinking water showing that they were potentially drawing on their own interactions with water across daily life, but not distinguishing water processes indoors versus out. This confusion was dominantly evidenced in Model 1 when students' understanding of the built environment was low overall. Of the 18 students who showed muddled indoor/outdoor water concepts in Model 1, only four students maintained this confusion in Models 2 and 3.

### 3.3.2. Missing Drainage Elements

Many student drawings were missing gutters as one of the key means that water is transported from the roof to the ground. This connection was mostly missing in the first models, and many students commented in the self-evaluation that they learned more about drains and pipes between Models 1 and 2. In other instances, it was visible in the drawings that students were adding more infrastructure to subsequent models based on their new knowledge.

### 3.3.3. Roofs Water Plants Below

When asked "why this process matters" (see File S1 modeling packet in Supplementary Materials), it was common for students to write about the benefit of water coming off roofs to help plant life below the building. There was thus an assumption made by numerous students that the water that falls off roofs is automatically good for plants below. This error may have been related to the missing drainpipes theme, where students envisioned rain falling off the roof toward the ground without the intentional design of buildings to move water toward sewers. While rain from roofs can sometimes help landscaping below, it was not clear that students understood that rainwater is commonly channeled directly into sewer systems without helping plants first.

### 3.3.4. Water "Just Hangs Out" on Roofs

Perhaps due to the emphasis on the water cycle, puddles, and evaporation, some students were inclined to show pools of water on flat roofs sitting there waiting to evaporate, a finding which relates to previous work with third-grade students [22]. In the current study, the finding was another indicator that students needed additional insight into infrastructure,

such as gutters and downpipes, and a better understanding that these elements occur on all buildings in climates with sufficient precipitation to necessitate gutters. This code was only tagged on models with conventional roofs, where the student did not draw or write about a green roof. Students who discussed water retention in relation to the green roof demonstrated one of the key learning outcomes of the unit.

### 3.3.5. The Water Cycle Requires Roofs

Some students went so far as to imply or state that the water cycle cannot happen without the roof. The idea of buildings related to the water cycle was a novel connection for students, as evidenced by the low instance of built environment and combined themes in Model 1 (Figures 5–7). However, upon adding the concept of "roof" to their mental model of the water cycle, some students overstated the importance of the roof. For example, Lucy wrote about her Model 3 drawing by explaining that "the water gets in the plants, then it cleans the air, my roof also has a pipe that puts the water in the street" and that this is important because "then the water cycle won't happen".

### 3.3.6. Epic Consequences

As students progressed through the unit, they were introduced to the importance of managing stormwater in cities to avoid flooding and destruction, and they appeared to be building on water cycle knowledge from previous years that helped them to understand the importance of water for life on earth. Students who emphasized disaster and dire consequences in their models were thus not exhibiting an inaccuracy but rather a tendency to exaggerate the broader impacts of these themes on nature or human communities. Most of the excerpts tagged with this code alluded to the widespread death of plant and/or human life, such as Sam, who said, "without all this good happy water, our survival probably would fail", and Addie, who says that, "everything would be dead" without these processes.

Teacher data provides additional insight into the inaccuracies identified in student models. To begin, in the pre-unit focus group, teachers at both elementary schools mentioned experience teaching the water cycle but that the concept of green roofs was new to them. When asked about their existing knowledge of green roofs, Mary said, "it's very new", and Gina added, "I know they [green roofs] exist, and that's about all I know". The teachers were, therefore, better positioned to teach the water cycle portion of the unit but needed additional time and resources to guide the green roof lessons within the unit. Post-unit, teachers also noted the effectiveness of the experiential learning activities, particularly the "rain to drain" lesson where students identified drainage elements on their own school building. This activity may have supported students in adding drainage elements to their drawings after Model 1. The teachers also noted that, overall, they desired more time to enact the unit and expressed that some lessons felt rushed because other lessons took longer than expected. This lack of time may relate to student confusion in making clear connections between green roofs and the water cycle at the end of the unit.

## 4. Discussion

This study examined the evolution of fourth-grade student ideas across three points in time across a 4-week curricular unit called "Raising the Green Roof". Prior work has begun to explore secondary students' understanding of complex systems over time [49–51]. However, there is little work examining primary students' beginning forays into building complex system thinking processes. The analyses of 212 systems models for 73 unique students allowed for the examination of how and if fourth-grade students were integrating knowledge across socio-hydrologic systems (natural and built environments) to conceptualize the interrelationships between the water cycle and buildings. The systems model data allowed a portal into students' spatial concepts expressed in the drawings paired with drawing annotations and writings that enhanced the researcher's interpretation of drawing

content. Focus groups with teachers, including a mid-analysis member check, assisted the research team in the sense-making of the student artifacts.

We used a recently proposed water literacy model [11] and were able to ground this theoretical model within our data. We found that previous conceptualizations of water literacy were a good fit for the analysis of student models. In particular, the McCarroll and Hamann [11] framework guided the view of student understanding as general versus specific, where student models showed a clear distinction between learners who demonstrated a broad level of understanding (e.g., green roofs are good for the water cycle) versus learners who were increasingly able to show specificity in their models (e.g., green roofs slowing down stormwater and helping cities better manage water flow).

The McCarroll and Hamann [11] framework included "hydrosocial knowledge", a topic that has been addressed by previous scholars examining water literacy across grade levels [22,52]. Previous studies have shown the importance of social themes in water cycle units, given that the water cycle is often taught in a way that is detached from humans and our built environments. For example, Abbott et al. [16] analyzed 464 water cycle diagrams from across the world and found that only 15% showed human interaction with the water cycle. Shepardson et al. [20] found that Midwestern students in grades 4–12 tended to exclude human interactions when prompted to draw and explain watersheds. In the current study, hydrosocial knowledge (as characterized by McCarroll and Hamann [11]) was addressed in the "Problems" module primarily in the context of flooding in cities, which grounded the water literacy unit in a social context. However, fourth-graders went well beyond flooding to readily connect unit themes to an array of human well-being considerations (see Table 2). Student integration of social dimensions was most evident in the ways that students connected the presence of plants to oxygen and clean air. Students also made facile connections between rain events, built environment, and flooding in cities. This work builds on findings at the secondary level [50,51] suggesting that as students were able to make their mental models visible of both systems, they were able to consider the relationships that occur between the systems (i.e., as shown in the idealized model progression in Figure 3). This work further demonstrates the possibility that water literacy units can provide socio-hydrologic framing to assist students in more readily connecting human and ecological systems.

This work joins previous studies that elucidate the ways in which complex systems can be seeded at the elementary level [10,53] to give a comprehensive understanding of water-human systems [13]. Our results suggested that the fourth-grade students were able to grasp foundational elements of the interrelationship between complex systems, showed increasing sophistication on elements within the systems, and articulated relationships between elements. As students learned about how a green roof interacts with the water cycle, they were able to build upon their simple causal reasoning structures to begin the shift toward complex causal reasoning.

To make this shift, students need opportunities to activate their simple causal reasoning structures [31]. Students demonstrated a high level of understanding of the water cycle in Model 1, which was prior to the unit, showing that many students held prior knowledge of the water cycle before starting the unit. Students also likely arrived with some knowledge of plants, given that plant biology is commonly integrated into U.S. early elementary education [7]. Prior knowledge about water and plants, therefore, provided the foundation onto which students could layer new understandings about the built environment and roofs. Once activated in Model 1, their ideas could be evaluated, refined, and/or built upon, which created opportunities for students to shift and integrate their ideas into more complex causal reasoning about the socio-hydrologic system [54,55]. As shown in the systems model analysis, students exhibited a variety of ways in which they refined and expanded their causal reasoning about the interconnections between the built environment and the water cycle.

The analysis of student confusion in the models complements the analysis of student knowledge demonstrations to draw a fuller picture of the student experience in the unit.

Much previous work has uncovered the confusions that students have surrounding the water cycle [18,19], such as confusing weather and climate, thinking that water only evaporates from oceans or lakes, and that raindrops always look like teardrops, to name just a few [21]. While there is much precedent literature on confusion surrounding hydrologic cycles, the inaccuracies identified in the current study centered primarily on the built environment. Curricular interventions at the intersection of K-8 STEM education and built environment are rare but increasingly available [40,56,57]. Given that built environment education [58] and green building literacy [9] have not been extensively studied, it is useful to better understand the ways in which elementary students need more support in these areas. This is especially the case for the development of curricular units that aspire to employ built environment ideas in STEM education.

*Implications for Practice & Research*

The current study demonstrated that elementary educators can provide the foundations for the integration of human and ecological systems. The unit evaluated here is built on learners' existing knowledge of the water cycle to consider how water moves in urban environments. The concept of green roof technologies was then layered into this baseline understanding as one potential solution for urban water cycle problems. Students were ultimately challenged to design their own doghouse by applying science content. Few science units are standards-aligned while making connections to everyday environments [59]. When successful, units such as "Raising the Green Roof" have the potential to provide the 3D learning experiences encouraged by the U.S. Next Generation Science Standards [7] that integrate multiple disciplinary core ideas (e.g., Earth science, human impacts, and engineering design) into one unit. The integration of built environment features, such as green roof technologies, has the additional benefit of fostering urban environmental education in, about, and for green infrastructure [60]. This means that students can learn the science of green infrastructure while making place-based connections to real projects in their communities. The hopeful long-term outcome is the cultivation of citizens positioned to advocate for green projects in their own cities. Another possibility for a curriculum rooted in engineering and design is that it can spark student interest in future green careers.

The current study revealed several practical implications for units such as "Raising the Green Roof". First, student and teacher data revealed the importance of learning that was both experiential and place-based (connected to the student's own school building), suggesting that some of the student learning successes evidenced in the data likely relate to the engaging and highly interactive design of the unit itself. Second, we discovered that built environment knowledge was low for elementary students, and additional support may be required to help both teachers and students learn about building features. The "Raising the Green Roof" unit, for example, would have benefited from more early emphasis on the flow of water off a typical roof to counter possible student confusion about how water interacts with buildings. Additional support may include engaging the facility manager of the school building if students are exploring their own school building and may also extend to guest lectures from local designers and engineers.

This curricular unit foregrounded the water cycle intersection with green roofs through the lens of improving human life in cities. While human health was not the main focus of the science content in the unit, we found that fourth-grade students naturally integrated "hydrosocial knowledge" [11] into their systems models. This concept could be enhanced in future unit implementations and provide the basis for future research. Increasing the hydrosocial dimensions could increasingly align the unit with ambitions to teach socio-scientific issues in the science classroom [53] and promote green building education [9]. This could include discussions of varying socio-economic contexts to question who experiences and benefits from green roof technologies.

Future implementations could additionally examine the benefits of green roofs beyond stormwater management to include topics such as urban heat islands, cleaner air, and habitat consideration [5], expanding the potential for green roofs to support other

domains of environmental literacy. Dry climates with little rainfall are less likely to have buildings with gutters or issues with stormwater management. The unit would, therefore, require adaptations for these locations. An adapted unit could address climate differences and the appropriateness and applicability of the green roof solution in varying climates (e.g., climates requiring drought-tolerant plantings to minimize potable water use). Further, the current unit was designed to contrast urban and rural settings and ultimately focus on stormwater management in cities. Rural educators could examine and adapt content for non-urban settings where flood mitigation is likely less of a concern.

## 5. Conclusions

Green building concepts cover far-ranging scientific concepts such as energy, water, materials, and beyond [61]. As such, the science of green building provides strong potential connections to lessons taught in K-12 science classrooms. Efforts to provide "Green Building Literacy" [9] to youth and the general public seek to address this gap. Green roof technologies comprise one of many possible green building strategies that could be integrated into the science classroom, where the current work examines the complex interrelationships between the water cycle, building roofs, and surrounding urban environments.

The "Raising the Green Roof" unit was designed with the ambition to teach complex systems and engage fourth-grade students in the ways in which the built environment affects the water cycle. The results of this study showed that students demonstrated increasing specificity in their models over time and were increasingly able to connect human and ecological systems in both their drawings and writings. We also discovered that built environment themes should be introduced in the unit better, and the effectiveness of the "rain to drain" activity showed promise for the use of the school building itself as a teaching tool for environmental education.

Implementing and evaluating units such as "Raising the Green Roof" can lead to an evidence-based curriculum for enhancing water literacy. The current unit made novel connections between the water cycle and green roof technologies, laying the groundwork for water literacy and the interrelationships between human and natural systems. The approach goes beyond the conventional approach to teaching the water cycle that often remains in the ecological sphere disconnected from human-built environments. Integrating the water cycle with human-built cities offers exciting ways to ground student learning in their nearby environments. It additionally provides a lens through which to see buildings and landscapes anew through the viewpoint of water flowing between interconnected systems.

**Supplementary Materials:** The following supporting information can be downloaded at: https://www.mdpi.com/article/10.3390/su16104262/s1, Figure S1: "Raising the Green Roof" Images from Unit Enactment; Figure S2: "Raising the Green Roof" Data Analysis Process; File S1: Student Modeling Packet.

**Author Contributions:** Conceptualization, L.B.C., L.P., L.Z., B.K.-G. and J.B.; methodology, L.B.C., L.P. and L.Z.; formal analysis, L.B.C., L.P., L.Z. and B.K.-G.; investigation, L.B.C., L.P., L.Z. and B.K.-G.; resources, L.B.C., L.Z., B.K.-G. and J.B.; data curation, L.B.C. and L.P.; writing—original draft preparation, L.B.C., L.P. and L.Z.; writing—review and editing, L.B.C., L.P., L.Z., B.K.-G. and J.B.; visualization, L.B.C. and L.P.; supervision, L.B.C.; project administration, L.B.C.; funding acquisition, L.B.C., L.Z., B.K.-G. and J.B. All authors have read and agreed to the published version of the manuscript.

**Funding:** This research was generously funded by the University of Missouri System Strategic Investment Program Tier 3.

**Institutional Review Board Statement:** The study was conducted in accordance with the Declaration of Helsinki and approved by the Institutional Review Board of the University of Missouri (project #20432442, approved December 2020).

**Informed Consent Statement:** Informed consent was obtained from all subjects involved in the study.

**Data Availability Statement:** The datasets generated and analyzed during the current study are not publicly available due to the need to protect the youth participants. Deidentified data are available from the corresponding author on reasonable request.

**Acknowledgments:** The authors would like to thank the teachers and students who enthusiastically adopted and participated in the "Raising the Green Roof" unit. We would additionally like to acknowledge and thank the following graduate students at the University of Missouri who supported this work in various stages of development: Sepideh Fallahhosseini, Suzanne Otto, Rebekah Snyder, and Mohammad Dastmalchi. We also extend our gratitude to Suzie Linihan at Colorado State University, who assisted with the inter-rater reliability of our qualitative data.

**Conflicts of Interest:** The authors declare no conflicts of interest.

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
