# Peer review of "Raising the Green Roof: Enhancing Youth Water Literacy through Built Environment Education"

_sustainability, doi:10.3390/su16104262_

Round 1

Reviewer 1 Report

Comments and Suggestions for Authors

The paper offers valuable insights into the role of built environment education in enhancing youth water literacy, with a novel approach to integrating green roof technology into elementary education.  To enhance the paper's clarity, impact, and applicability, it is recommended that the authors make modifications to address the aforementioned deficiencies in following.  Specifically, refining the structure and clarity of the methodology, expanding on the innovative potential of the educational model, and bolstering the validity and data support of the findings will significantly improve the paper's contribution to the field of sustainability education.

1. The paper is generally well-written and clear, though some technical terms and concepts could be better defined for readers unfamiliar with education theory or green infrastructure. Include a glossary or define key terms when first mentioned to improve accessibility for a broader audience. Change "systems models" to "systems models (diagrams that represent interconnected environmental and human-made processes)".

2. The study's approach to enhancing water literacy through green roof education is innovative, yet the connection to broader environmental literacy and potential scalability of the intervention could be explored further. Discuss the potential for applying this educational model in diverse geographic and socio-economic contexts, including urban and rural settings. Expand the discussion to include potential adaptations of the curriculum for different climates and types of green infrastructure beyond green roofs.

3. The research methodology is solid, but the paper could provide more detailed information on the controls in place to ensure the validity of the student assessments and the reproducibility of the study. Elaborate on the measures taken to avoid bias in the evaluation of students' models and written reflections. Describe the training process for researchers or educators assessing student work to ensure consistency in the evaluation criteria.

4. While the paper presents compelling qualitative data, the integration of quantitative data to support findings could further strengthen the argument. If available, include statistical analysis of pre- and post-intervention assessments to quantitatively demonstrate changes in students' understanding of water systems. Add a section on statistical analysis comparing the number of students who could accurately describe the water cycle and green infrastructure's role before and after the intervention.

Author Response

Thank you for your constructive feedback on our manuscript. We have integrated reviewer feedback and believe the manuscript is much improved from the last submission. Please see point-by-point responses to your concerns below.

Reviewer 1 

The paper offers valuable insights into the role of built environment education in enhancing youth water literacy, with a novel approach to integrating green roof technology into elementary education.  To enhance the paper's clarity, impact, and applicability, it is recommended that the authors make modifications to address the aforementioned deficiencies in following.  Specifically, refining the structure and clarity of the methodology, expanding on the innovative potential of the educational model, and bolstering the validity and data support of the findings will significantly improve the paper's contribution to the field of sustainability education. 

  1. The paper is generally well-written and clear, though some technical terms and concepts could be better defined for readers unfamiliar with education theory or green infrastructure. Include a glossary or define key terms when first mentioned to improve accessibility for a broader audience. Change "systems models" to "systems models (diagrams that represent interconnected environmental and human-made processes)".

We have re-read the paper searching for jargon that needs better definition when first introduced. We have added clarity for stormwater infrastructure and system models as suggested. We have also substantially re-written Section 1.1.4 on Model-based learning to give better definitions for the educational concepts.

  1. The study's approach to enhancing water literacy through green roof education is innovative, yet the connection to broader environmental literacy and potential scalability of the intervention could be explored further. Discuss the potential for applying this educational model in diverse geographic and socio-economic contexts, including urban and rural settings. Expand the discussion to include potential adaptations of the curriculum for different climates and types of green infrastructure beyond green roofs.

We appreciate these comments and have expanded our discussion surrounding curricular expansions and improvements in section 4.1 “Implications for Research and Practice.” 

  1. The research methodology is solid, but the paper could provide more detailed information on the controls in place to ensure the validity of the student assessments and the reproducibility of the study. Elaborate on the measures taken to avoid bias in the evaluation of students' models and written reflections. Describe the training process for researchers or educators assessing student work to ensure consistency in the evaluation criteria.

The 2.4 Data Analysis section has been enhanced with validity measures. We have added the results of our inter-rater reliability tests. On p.11, we have also added a note about the training of graduate research assistants prior to data coding. We have also added a note about the interdisciplinary composition of the research team as a way to reduce disciplinary bias.

  1. While the paper presents compelling qualitative data, the integration of quantitative data to support findings could further strengthen the argument. If available, include statistical analysis of pre- and post-intervention assessments to quantitatively demonstrate changes in students' understanding of water systems. Add a section on statistical analysis comparing the number of students who could accurately describe the water cycle and green infrastructure's role before and after the intervention.

Since its inception, we have been conceptualizing this paper as an in-depth qualitative analysis. However, we can see how presenting code frequencies would lead to reader expectations for statistics. We have now added results that include the McNemar test for differences with pre/post dichotomous, categorical data. These results are now integrated into the results section. See section 3.2 “How was Socio-hydrologic Systems Thinking Expressed throughout the Unit?”

Reviewer 2 Report

Comments and Suggestions for Authors

I would like to point out several structural issues that significantly hinder its readability and comprehension. Specifically, the organization of content within the Results section includes explanations of concepts that would be more appropriately situated in the Materials and Methods section. This misplacement disrupts the logical flow of information and may confuse readers seeking to understand the foundational elements of your study before delving into its outcomes.

Incorporating a flow diagram would not only enhance the interpretability of the methodology employed but also contribute to a clearer separation and definition of the manuscript's various sections.

Figure 2. Water Literacy Framework (McCarroll and Hamann, 2020) (Source: Permission to use granted from authors)

Figure 2 is identical to a figure from another paper, which is available in an open access journal. Even if permission from the original authors has been obtained, I must emphasize that republishing the same figure without any modifications does not contribute new information or value to the current paper. Therefore, I strongly recommend either significantly altering the figure to highlight new insights relevant to your findings or focusing on creating original graphics that support your narrative directly. Additionally, it is crucial to adhere to the journal's citation standards without exception. As the article frequently refers to the aforementioned graph, it will be necessary to remove those references and replace them with descriptions of the concepts that are independent of their color or position within the graph.

Upon closer examination of Figure 1, which is described as an adaptation, it appears to be a direct copy of an existing graph. Although the iconography has been changed, the information presented remains unchanged. This suggests that the figure is more of a reproduction than a true adaptation. As such, it fails to contribute any new insights or value to the paper. It is crucial for the integrity of academic work that all figures and graphical representations genuinely advance the understanding of the topic being discussed. Simply altering the visual style without adding new content or perspective does not meet the standards of originality and contribution expected. I recommend revising or replacing this figure with one that provides unique insights or information pertinent to your research.

The current format of the citations does not align with the citation style guide specified by journal.

Line 177 “humans…dominate critical components of the hydrosphere”. This sentence needs a bibliographic citation.

Regarding Figure 8, it appears that the results from models 2 and 3 are combined, which could potentially obscure the progression and comparative analysis of these models. If there is a sequential or developmental relationship between these models, separating their results into distinct parts or figures would provide a clearer representation of their respective outcomes and differences.

It's pertinent to note that the focus of the results section appears to be overly centered on the individual analysis of children's drawings. While such detailed examination might hold value within the context of teaching methodology recommendations, it does not align with the expectations for a scientific article. The essence of scientific research lies in deriving broader insights that contribute to the field's understanding, which in this case, would mean orienting the analysis towards more global, generalized findings rather than isolated instances.

Furthermore, regarding the observation of plastic roofing on cardboard boxes being mistaken for vegetative cover due to its green color, this highlights a missed opportunity to engage children with nature directly. Incorporating natural elements from the immediate environment, such as ruderal plants, annuals, or fast-germinating seeds, could foster a deeper connection with and understanding of ecological principles. While integrating genuine natural elements into the activity might present additional challenges, it aligns more closely with the educational objectives aimed at instilling environmental sensibilities in children. This approach not only enriches the educational experience but also ensures that it resonates with the principles being taught, thereby enhancing the scientific validity and impact of the study.

Author Response

Thank you for your constructive feedback on our manuscript. We have integrated reviewer feedback and believe the manuscript is much improved from the last submission. Please see point-by-point responses to your concerns below.

Reviewer 2 

I would like to point out several structural issues that significantly hinder its readability and comprehension. Specifically, the organization of content within the Results section includes explanations of concepts that would be more appropriately situated in the Materials and Methods section. This misplacement disrupts the logical flow of information and may confuse readers seeking to understand the foundational elements of your study before delving into its outcomes. 

We can see how our results section, optically, appears wordy. This paper is written in the qualitative tradition of weaving together data sources. That said, based on this comment, we reviewed our results section for confusing passages. We did find that we were over-explaining or unnecessarily re-stating our methods. We have omitted these passages or moved them up into the methods section. We appreciate this comment that helped us to streamline the presentation of results.

Incorporating a flow diagram would not only enhance the interpretability of the methodology employed but also contribute to a clearer separation and definition of the manuscript's various sections. 

We agree that a flow diagram can help the reader follow the multi-phase analytical process. We have created and included a new flow diagram that shows the research process from data collection to data analysis. See supplemental materials Figure S3.

Figure 2. Water Literacy Framework (McCarroll and Hamann, 2020) (Source: Permission to use granted from authors) Figure 2 is identical to a figure from another paper, which is available in an open access journal. Even if permission from the original authors has been obtained, I must emphasize that republishing the same figure without any modifications does not contribute new information or value to the current paper. Therefore, I strongly recommend either significantly altering the figure to highlight new insights relevant to your findings or focusing on creating original graphics that support your narrative directly. Additionally, it is crucial to adhere to the journal's citation standards without exception. As the article frequently refers to the aforementioned graph, it will be necessary to remove those references and replace them with descriptions of the concepts that are independent of their color or position within the graph. 

We have removed the original Figure 2 from the current manuscript and left the reference to the McCarroll and Hamann paper where interested readers can find the figure. We adapted our narrative accordingly.

Upon closer examination of Figure 1, which is described as an adaptation, it appears to be a direct copy of an existing graph. Although the iconography has been changed, the information presented remains unchanged. This suggests that the figure is more of a reproduction than a true adaptation. As such, it fails to contribute any new insights or value to the paper. It is crucial for the integrity of academic work that all figures and graphical representations genuinely advance the understanding of the topic being discussed. Simply altering the visual style without adding new content or perspective does not meet the standards of originality and contribution expected. I recommend revising or replacing this figure with one that provides unique insights or information pertinent to your research. 

We omitted this figure and left the reference to the website where an interested reader can find the information.

The current format of the citations does not align with the citation style guide specified by journal. 

We have aligned all citations with the guidelines provided by the journal.

Line 177 “humans…dominate critical components of the hydrosphere”. This sentence needs a bibliographic citation. 

The citation with page number was at the end of the sentence. We moved it directly behind the quote so that readers don’t miss it.

Regarding Figure 8, it appears that the results from models 2 and 3 are combined, which could potentially obscure the progression and comparative analysis of these models. If there is a sequential or developmental relationship between these models, separating their results into distinct parts or figures would provide a clearer representation of their respective outcomes and differences. 

We agree that this muddies the analysis. We have gone back to the data and isolated the model counts for Model 3. Given the request to provide statistical analyses pre-post by Reviewer 1, we have omitted Model 2 counts from the paper and focused on the differences pre/post. The new figures (Figures 6-7) have been added/improved to align with the additional analysis and highlight changes over time. We hope that the data is better visualized with these changes.

It's pertinent to note that the focus of the results section appears to be overly centered on the individual analysis of children's drawings. While such detailed examination might hold value within the context of teaching methodology recommendations, it does not align with the expectations for a scientific article. The essence of scientific research lies in deriving broader insights that contribute to the field's understanding, which in this case, would mean orienting the analysis towards more global, generalized findings rather than isolated instances. 

This research was conducted using a qualitative social scientific approach to elucidate student scientific understandings of socio-hydrologic systems. We would like to emphasize that our sample (73 students, 212 unique models) is a large sample size for qualitative work. We found that descriptive statistics were a strong way to represent our large dataset, and that frequences of codes could be examined statistically pre-post to highlight significant changes over time. However, a consensus-based, qualitative analysis is at the heart of this paper and the process used to treat our rich data that included student drawings and writings along with teacher perspectives. The revised paper has a strengthened section on data analysis as prompted by reviewer comments.

The presentation of individual student drawings is reported after the presentation of the larger trends across the dataset and these individual drawings are meant to bring life to our data. They show the reader model progressions that typify our analytical categories. We believe that the paper would be lacking if our readers did not have access to the student artifacts. However, we would be willing to shorten the number of examples or move them to supplemental materials if the editorial team prefers this approach.

Furthermore, regarding the observation of plastic roofing on cardboard boxes being mistaken for vegetative cover due to its green color, this highlights a missed opportunity to engage children with nature directly. Incorporating natural elements from the immediate environment, such as ruderal plants, annuals, or fast-germinating seeds, could foster a deeper connection with and understanding of ecological principles. While integrating genuine natural elements into the activity might present additional challenges, it aligns more closely with the educational objectives aimed at instilling environmental sensibilities in children. This approach not only enriches the educational experience but also ensures that it resonates with the principles being taught, thereby enhancing the scientific validity and impact of the study. 

The unit had numerous place-based engagements with plants. Students worked with real plants to learn about plant transpiration, students toured their school grounds examining permeable surfaces with jugs of water, and they experienced a live broadcast from an engineer on a green roof. If the unit had not occurred during COVID, a field trip could have been possible. We have improved our description of the unit to highlight the hands-on activities students did. We also mentioned future possibilities for improvement in the discussion section.